# Deployment of Smart Specimen Transport System Using RFID and NB-IoT Technologies for Hospital Laboratory

**DOI:** 10.3390/s23010546

**Published:** 2023-01-03

**Authors:** Ngoc Thien Le, Mya Myet Thwe Chit, Thanh Le Truong, Atchasai Siritantikorn, Narisorn Kongruttanachok, Widhyakorn Asdornwised, Surachai Chaitusaney, Watit Benjapolakul

**Affiliations:** 1Center of Excellence in Artificial Intelligence, Machine Learning and Smart Grid Technology, Department of Electrical Engineering, Faculty of Engineering, Chulalongkorn University, Bangkok 10330, Thailand; 2Department of Laboratory Medicine, Faculty of Medicine, Chulalongkorn University, Bangkok 10330, Thailand

**Keywords:** RFID, passive RFID tag, NB-IoT technology, specimen tube, specimen transport box

## Abstract

In this study, we propose a specimen tube prototype and smart specimen transport box using radio frequency identification (RFID) and narrow band–Internet of Things (NB-IoT) technology to use in the Department of Laboratory Medicine, King Chulalongkorn Memorial Hospital. Our proposed method replaces the existing system, based on barcode technology, with shortage usage and low reliability. In addition, tube-tagged barcode has not eliminated the lost or incorrect delivery issues in many laboratories. In this solution, the passive RFID tag is attached to the surface of the specimen tube and stores information such as patient records, required tests, and receiver laboratory location. This information can be written and read multiple times using an RFID device. While delivering the specimen tubes via our proposed smart specimen transport box from one clinical laboratory to another, the NB-IoT attached to the box monitors the temperature and humidity values inside the box and tracks the box’s GPS location to check whether the box arrives at the destination. The environmental condition inside the specimen transport box is sent to the cloud and can be monitored by doctors. The experimental results have proven the innovation of our solution and opened a new dimension for integrating RFID and IoT technologies into the specimen logistic system in the hospital.

## 1. Introduction

Nowadays, radio frequency identification (RFID) and Internet of Things (IoT) technology play a fundamental role in the so-called automation factory or Industry 4.0 era, aiming to increase the level of automation of industrial processes. In the medical packaging and logistic domain, RFID has gradually become an attractive technology due to its distinctive features such as the low cost of RFID tags, the reuse of RFID tags, and the easiness of RFID tag’s deployment and within the items to be tracked [1,2,3,4,5,6,7,8,9,10]. In addition, RFID and IoT are vital to boost the hospital’s digital transformation to improve the efficiency and safety through many applications, such as tracking patients inside the hospital, drug management, and medical assets in hospitals. Despite the rising RFID and IoT applications in medicine, there is a limited medical specimen management and logistic domain approach for hospitals [11,12,13,14,15]. In fact, medical specimen tube management is one of the critical activities in hospitals with extreme accuracy and reliability in the labeling, delivery, and recording processes. Any errors have significant consequences for patient care and healthcare management and increased costs are often unaccounted for by patients.

The motivation behind our work is to solve the existing specimen transport system and management problems at King Chulalongkorn Memorial Hospital. This hospital is one of the largest public hospitals in Bangkok, Thailand, with an in-patient capacity of over 1400 beds. The number of patients admitted in 2017 was 1.6 million outpatients and more than 5000 patients daily. Blood or other samples such as urine, feces, sputum, pus, skin, hair, and nails are collected from the patients as required by the doctors. Then, these specimens must be transported to the Central Laboratory, Department of Laboratory Medicine, for examination. Unfortunately, the document-based specimen transport system cannot handle a massive request for specimen delivery while minimizing human errors such as mislabeling specimens, losing tubes, or incorrect delivery. Furthermore, the Central Laboratory also receives the specimen tubes from other local hospitals since they do not have sufficient equipment for examination. This specimen tube is encased in the transport box with an appropriate temperature and humidity requirement during transportation. In addition, the Central Laboratory must be able to remotely monitor the temperature and humidity of the box and trace the box location to estimate the delivery time. Therefore, it is necessary to develop an alternative specimen transport system utilizing digital technologies to replace the document-based one for the King Chulalongkorn Memorial Hospital.

The barcode solution for specimen tubes in the literature cannot be a solution to the above problems. The barcode technology requires reading data in the line-of-sight only one label at a time; hence, the specimen tube processing rate is much slower than RFID technology. In fact, the biggest drawback of the barcode approach is that the barcode falls short of usage because the printed code is easily damaged or washed out. Moreover, another requirement is that the tubes in the transfer box must be placed under suitable temperature and humidity conditions while being delivered from one laboratory to another.

Our work solves the above problems of specimen transport system at the King Chulalongkorn Memorial Hospital by introducing a smart specimen transport system using RFID and NB-IoT technologies. The passive RFID tags can be multiple-read and write information even under non-line-of-sight conditions. The tags are made with plastic so that they are waterproof and heatproof. Furthermore, an RFID tag can easily be attached to specimen tubes requiring sterilization. Additionally, the NB-IoT device for the specimen transport box monitors temperature and humidity values inside the box during transportation. Moreover, the GPS location is also reported to the cloud platform to estimate the delivery duration. By introducing the smart specimen transport system, our work also contributes to widening the innovative applications of RFID and IoT technologies used in the clinical laboratory environment. In conclusion, the key features of our proposed system are:The multiple reading/writing RFID system is located at a laboratory for traceability and accounting for the specimen tubes during the delivery to hospital laboratories;The portable RFID reader attached to the transfer box to check the correct receiver laboratory for specimen tubes;The NB-IoT device for tracking the specimen box position and monitoring temperature, and humidity inside the box.

The rest of the paper is organized as follows. Section 2 describes the literature review of RFID and IoT applications in hospital management and logistics supply chains. In Section 3, we first introduce the multi-tag RFID system for specimen tubes. Then, the IoT-based monitoring system is introduced with the processing unit and the backend server on the cloud via the NB-IoT network. The resulting experiments of our proposed method are analyzed in Section 4. Finally, we make our concluding remarks in Section 5.

## 2. Related Work

RFID technology has been widely applied in worldwide management and logistics supply chains. There are two scenarios of RFID usage, namely under indoor and outdoor conditions [2,16,17,18]. For indoor RFID usage, the users utilized RFID tags to manage many resources such as patient healthcare information, medical equipment, or drug delivery in the hospital. For outdoor RFID usage, the RFID tags are usually used for storing information and tracing medical items such as medicine, blood, and specimens during traveling. In addition, IoT technology is also utilized for those outdoor scenarios to monitor the environmental condition or to track an item’s location.

The authors in [19] implemented the HF range RFID tags (13.56 Mhz) for the specimen management system in Yonsei University Wonju Christian Hospital for the indoor clinical laboratory. Instead of barcodes and stickers, the RFID tags containing patient information are stuck on the specimen holders. Their proposed RFID system could also write and read data at up to 20 tags per second. Other typical RFID indoor usages are a patient’s tracking and localization system, medical staff, nurse, and their interaction management [20,21,22,23]. In detail, the HoMeTrack system in [20] can track the route and timeline of medicine out of the storage area until they arrive at the patients. The proposed approach in [21] is also helpful for managing expensive equipment, surge equipment in the surge room, or bedside equipment. Unlike [20,21,22], the proposed RFID systems in [23] used the active UHF tags (916.5 MHz) in localization service and medicare applications. A demonstration of RFID application in Tokyo Medical and Dental University [24] has successfully used both active and passive RFID tags to track the clinical intervention and blood tests of the patients together with the wireless communication system. RFID tags have also been implemented for the laboratory specimen management to replace the barcode system [11,12,13,14,15]. For example, the RFID systems in [14,15] were proposed to manage and authenticate blood samples in hospital clinical laboratories. In general, the most significant advantage of these studies is that of mass information processing using RFID tags.

The development of IoT technologies has create new spaces for the integration of RFID tags for outdoor usage [4,5,6,7,8,9,10,25,26,27,28]. In [25], the authors proposed tracking and tracing food conditions using the integration of RFID and IoT technologies. The IoT is integrated into the RFID gateway and transfers the environmental conditions and data stored on RFID tags via WiFi. In a smart healthcare system in [26], the authors proposed NodeMCU and Blynk software as IoT platforms to track and monitor the patient’s vital signals. The patient information is recorded in an RFID tag and transferred to the hospital based on the WiFi connection of the IoT. In [27], the authors utilized RFID and IoT technologies to monitor the temperature of the cold medicine chain. Their proposed IoT system was based on a wireless sensor network, and the RFID tag was used to record medical information. From the literature, although there are many outdoor applications of RFID and IoT technologies for medical cold chain logistics, such devices lack a specimen transport system in the market. Some of the available commercial products of specimen boxes, such as the LABCOLD device [29] or Thermoelectric cooler [30] (Figure 1), are not equipped with RFID and IoT technologies to enable the monitoring of environmental conditions or tracing the position of the box during traveling.

## 3. Implementation of Proposed Smart Specimen Transport Box

Figure 2 illustrates the overall picture of our proposed specimen transportation method at the Department of Laboratory Medicine Chulalongkorn hospital. The RFID devices are installed in the laboratory rooms (with an external antenna) and the transfer box (with an on-board antenna). The NB-IoT device is installed at the specimen transfer box to send the temperature, humidity, and GPS data to the Magellan cloud service. The purpose of cloud storage is that the environment changes inside the transfer box can be monitored from both sender and receiver labs. The data link of our NB-IoT device is provided by the AIS network. In the following subsections, we describe the technical details of the main modules of our proposed devices. The description of modules is shown in Table 1.

### 3.1. Multiple Reading and Writing RFID Solution

#### 3.1.1. EPC Gen 2 UHF RFID tag

Figure 3 shows the tag used in our project, namely the Electronic Product Code (EPC) Gen 2 860–960 MHz UHF RFID tag, which supports a multi-read technology tag with a reading rate of up to 150 tags per second. The Gen 2 tag takes full advantage of the higher UHF bandwidth by supporting the reading of multiple tags, which is not possible in older RFID technologies. RFID readers accomplish this by asking RFID tags to randomly select two numbers, the reader then reduces the first number incrementally until all tags are read. The second number creates the priority sequence for tags which select the same first number. The board has an adjustable power output from 0 dBm to 27 dBm, meaning that, with the correct antenna, the reading range can be up to 16 feet (4.9 m) or up to 2 feet with the onboard antenna [31]. The writing of tags is also possible at a 80 ms standard write. The tag’s price is affordable with $2 per set of five tags [32], which is cheaper than the RFID tags used in [25] which cost approximately $2.1 per tag.

In our proposed system, the RFID tag does not include explicit data, thereby creating privacy for patient information. Only the sender laboratory can obtain the personal information of the patient by using the code of the patient. Furthermore, only the environment and GPS location of the box are uploaded to the cloud via the NB-IoT module. Those are effective solutions for protecting patient information during traveling.

#### 3.1.2. SparkFun Simultaneous RFID Reader

An RFID reader is one of the key components of our proposed system. The reader transmits radio electromagnetic waves to the medium. When the transponder enters the read range, the RFID tag’s internal antenna draws energy from the reader and sends it to IC, which generates a signal back to the system. The reader detects a backscatter and interpreted it to the readable information. Readers also use RF waves to write new information on the tag. Usually, the reader connects with the computer via a cable to display the detected tag information using middle-ware. Some have a built-in onboard antenna to detect the tag in the short-range, while some external support antennas can be used for greater ranges. We usee the Sparkfun simultaneous RFID reader—M6E nano board [33].

#### 3.1.3. UHF External Antenna

The ultra high-frequency (UHF) RFID Antenna boasts a frequency range of 860–960 MHz with a gain of six dBi. This is the external antenna we use for our simultaneous RFID reader due to its high-quality features [34].

#### 3.1.4. Arduino Mega 2560

The Arduino device has been widely used for fast prototyping in IoT applications. We use the Arduino Mega 2560 device for the signal processing of specimen boxes due to its low-cost, portable, high flash memory, with more serial pins for serial communication devices, and a variety of reference resources [35].

### 3.2. IoT Solution for Monitoring

#### 3.2.1. Temperature and Humidity Sensor Module

The SHT10 sensor module was chosen to monitor both the temperature and humidity inside the transport box. It is a tiny, low power consumption device, and it is Arduino-compatible and easy to deploy. It precisely calculates the temperature and humidity values and interfaces with the Arduino device via I2C communication [36].

#### 3.2.2. GPS Module

GPS-EM506RE is a coin-sized module with a high-accuracy and low-power consumption device. It works stably with the Arduino device at 4800 baud rates under indoor and outdoor conditions [37].

#### 3.2.3. NB-IoT Module

The AIS NB-IoT shield is chosen for our project because it is currently available on the market, license band, and is still releasing updated features timely [38]. The shield is implemented by the Message Queuing Telemetry Transport (MQTT) protocol for monitoring purposes. The shield is an Arduino-compatible device that works with 9600 baud rates and supports the Magellan cloud platform. It uses an embedded e-SIM for cellular networks. On the Magellan IoT cloud platform, we can check the sensor data sent from the NB-IoT shield. Additionally, we can visualize the sensor data shown on the dashboard. Finally, Figure 4 shows all the components of specimen transport box in our study.

**Table 1 sensors-23-00546-t001:** Modules description.

No.	Module	Features	Ref.
1	Passive RFID tag	- EPC global Gen.2 and ISO/IEC 18000-6C - 800 bits memory - 512 bits user - 64-bit unique TID (unalterable serial number) - 32-bit access and 32-bit kill passwords - Width: 1 cm; length: 7 cm	[31]
2	Simultaneous RFID reader	- EPC global Gen. 2 (ISO 18000-6C) with normal backscatter rate of 250 Kbps - Separate read and write levels, command adjustable: 0 dBm to 27 dBm, steps 0.01 dB - 0.25 mW in shutdown mode - 0.84 W in ready mode - Up to 150 tags per second to read 96-bit EPC - 80 ms typical for standard write 96-bit EPC	[33]
3	UHF antenna	- Frequency range: 860–960 MHz - Gain: 6dBi; impedance: 500 ohm - Polarization: linear vertical; VSWR: 1.6:1 - Max power: 100 W; dimension: 223 mm × 200 mm × 60 mm	[34]
4	Arduino Mega 2560	- ATmega328 microcontroller - 14 digital I/O pins (6 PWM outputs) - 32KB flash memory (0.5KB for boot loader)	[35]
5	SHT10 sensor	- Interface type: serial (I2C non standard) - Humidity ranger: 0–100%RH - Temperature range: −40 °C−128 °C	[36]
6	GPS-EM506RE	- 48-channel receiver - Extremely high sensitivity: −163 dBm - − 2.5 m positional accuracy; 45–55 mA at 4.5–6.5 V	[37]
7	NB-IoT module	- NB-IoT module: Quectel BC95-B8 - Data transmission rate: 24 kbps (download); 15.625 kbps (upload) - NB-IoT protocol support: UDP, TCP, CoAP, IPv4, MQTT	[38]

## 4. Result Evaluation

In order to illustrate how our smart specimen transport box proposed in this contribution works, an experiment at the laboratory and a pilot test were conducted at the Central Laboratory of the Department of Laboratory Medicine, Chulalongkorn hospital Bangkok, Thailand. The laboratory experiment aimed to determine the suitable number of specimen tubes in each specimen box so that the RFID system could successfully read the information. The pilot test evaluates the NB-IoT’s performance in indoor and outdoor environments. The results are represented in the following subsections.

### 4.1. Performance of RFID System

#### 4.1.1. Determine the Number of Tubes Inside the Transport Box

Figure 5 illustrates our laboratory experiments at the Department of Laboratory Medicine. In the first experiment, we determine the optimal number of tubes placed in the transport box that can be read/written successfully by RFID with the external antenna. The power of RFID is set at 25 dBm and 27 dBm.

Currently, there are two options for the number of specimen tubes per transport box: 25 tubes and 50 tubes. Figure 6 shows our experiments with these two options. The total number of ice bags is ten in all cases. As in Figure 6, we repeat the measurement ten times for each test.

Table 2 shows the results of our first experiment. We found that 25 tubes per transport box are suitable for our RFID capability with a power of 25 dBm. In addition, the ice bag has a negative impact on our RFID system since if we place the ice bags on the front side (antenna side), the total number of detected tubes cannot reach 100% even with the maximum power of 27 dBm.

#### 4.1.2. Determine the Ice Bag’s Position Inside the Transport Box

The ice bag is necessary to transport the specimen tubes since it keeps them under cool conditions. However, the ice bag interferes with the RFID signal, as seen in the first experiment. Therefore, in the second experiment, the ice bag positions inside the box are varied to determine the appropriate position.

Figure 7 illustrates our experiments with 25 tubes in all cases. In addition, we fill water in all tubes in the last case, Figure 7D, to find a suitable distance between the box and external antenna that can successfully read/write by RFID technology. The power of RFID is fixed at 25 dBm. As in Figure 7, we repeat the measurement ten times for each test.

Table 3 shows the results of our second experiment. We found the suitable position for ice bags in Figure 7C,D. Therefore, in practice, we add an antenna label on the front side of the box so that the employee will not place ice bags on this side.

### 4.2. Performance of NB-IoT System

The NB-IoT prototype is integrated inside the specimen transport box to determine the box’s temperature, humidity, and location, as shown in Figure 4. The GPS receiver antenna is attached outside the transport box to enhance the satellite signal strength. In the pilot test, the specimen transport box receives 25 specimen tubes from the sender (laboratory A) and then goes to the receiver (laboratory B) located in a different building. Figure 8 illustrates our pilot test. According to the NASA technology readiness levels [39], our pilot test achieved a level 6–system adequacy validated in a simulated environment.

Furthermore, we developed a specimen tube destination indicator (STDI) device using the RFID reader attached to the transfer box with the onboard antenna. Each specimen tube must be delivered to a receiver’s clinical laboratory by recording the receiver laboratory’s GPS location to the RFID tag and following the check-in and check-out procedures, as shown in Figure 9.

In the actual situation, a transfer box contains many specimen tubes belonging to different receiver laboratories. Thus, any specimen tubes may be misplaced in the wrong receiver laboratory. This STDI device indicates the destination of a specimen tube by checking the GPS location of the receiver laboratory and compared to the current GPS location of the transfer box. If the absolute error between these two GPS locations is less than 10 m, we assume that they are the same location. One LED will blink when the specimen tube near the STDI device is placed. The green LED will blink if the specimen tube arrives at the corrected receiver laboratory. If a specimen tube belonging to a different receiver lab is taken out, the red LED will blink as an incorrect delivery indicator.

During transportation, the NB-IoT sends the temperature, humidity, and position data every minute to the Magellan Cloud. Therefore, the environmental conditions inside the transfer box and the location can be monitored. Additionally, the Magellan dashboard shows values in real-time during the box’s traveling, as shown in Figure 10. Finally, the comparison between our smart specimen transport system and other related works is given in Table 4 to highlight the innovation of our smart specimen transport system.

## 5. Conclusions

In this paper, we proposed the smart specimen transport box based on RFID and NB-IoT technologies to monitor the quality of the specimen tubes during traveling to accelerate the specimen accounting process and eliminate the lost specimens during delivery caused by employees. According to the experiments in the laboratory and pilot tests at Chulalongkorn hospital, our proposed system was represented as being practical. In addition, the passive RFID tag can be reused, which allows money to be saved. Such a system can be integrated into existing medical laboratory systems and open up new opportunities to eliminate the paper-based process and accelerate the medical information flow.

Future work will be simplified by the use of software in our proposed system making it more convenient for doctors to use the applied machine learning models to optimize the route for delivering transfer boxes and specimen tubes to many laboratories utilizing the traffic information. Another future work is that of implementing our smart specimen transport system in real clinical laboratories to evaluate the efficiency of the nurse and physician aspects. 

## Figures and Tables

**Figure 1 sensors-23-00546-f001:**
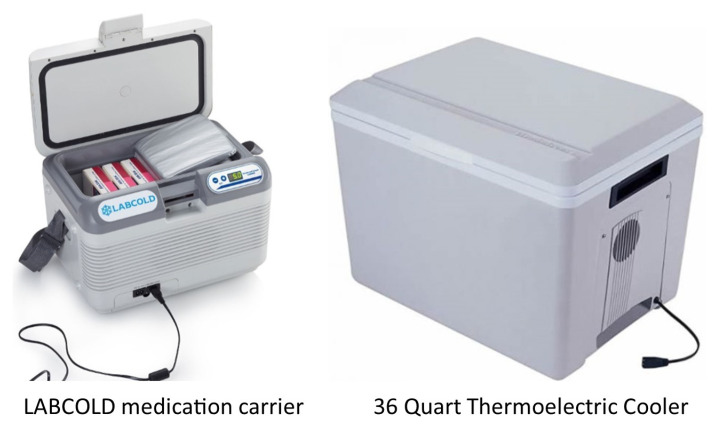
Examples of an existing specimen transport box for a hospital laboratory [29,30].

**Figure 2 sensors-23-00546-f002:**
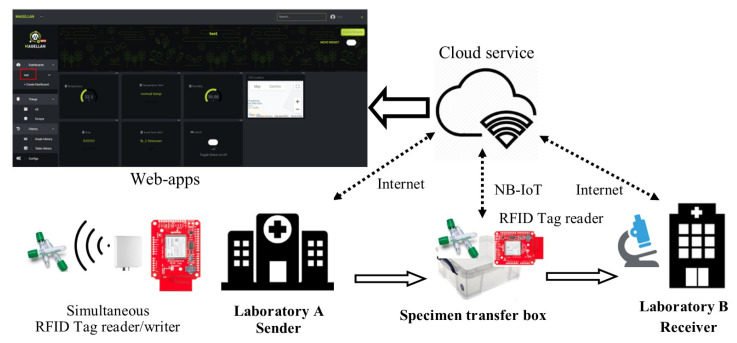
The overall proposed method for specimen transportation. Only the environment and GPS location of the box are uploaded to cloud via the NB-IoT module.

**Figure 3 sensors-23-00546-f003:**
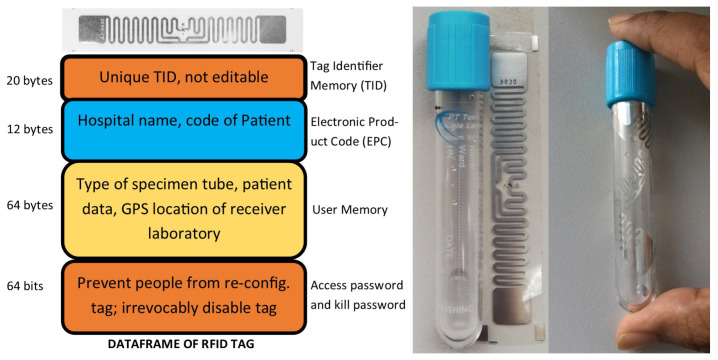
The passive RFID tag EPCglobal Gen2 used in our study [31]. The tag does not include explicit data, and thereby creates privacy for the patient information.

**Figure 4 sensors-23-00546-f004:**
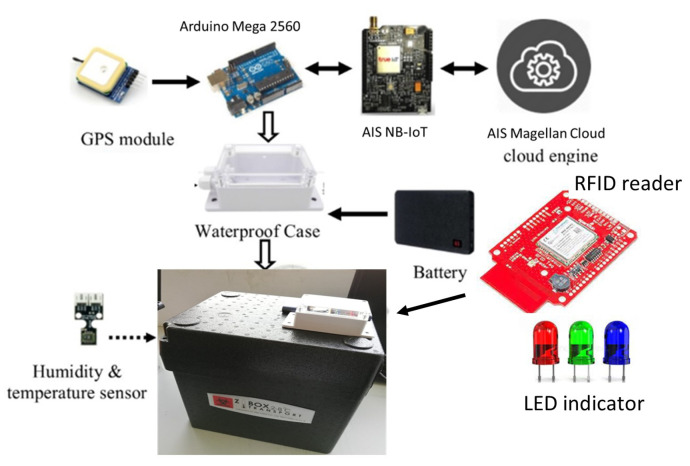
Components of specimen transport box. The box, commonly used at the Department of Laboratory Medicine, King Chulalongkorn Memorial Hospital, is from styrofoam, with a 5 cm thickness.

**Figure 5 sensors-23-00546-f005:**
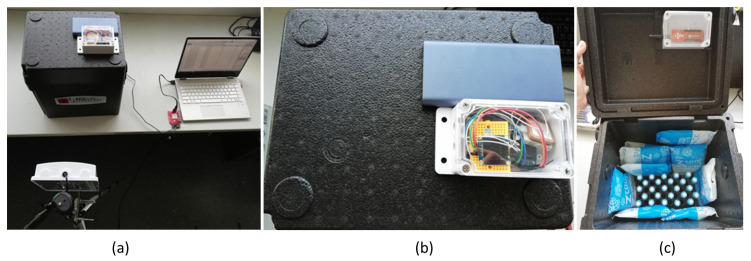
Our proposed RFID devices and smart specimen transport box using RFID and NB-IoT technologies. Overall proposed system (**a**), top-view of the specimen transport box (**b**), and tubes placed inside the box (**c**).

**Figure 6 sensors-23-00546-f006:**
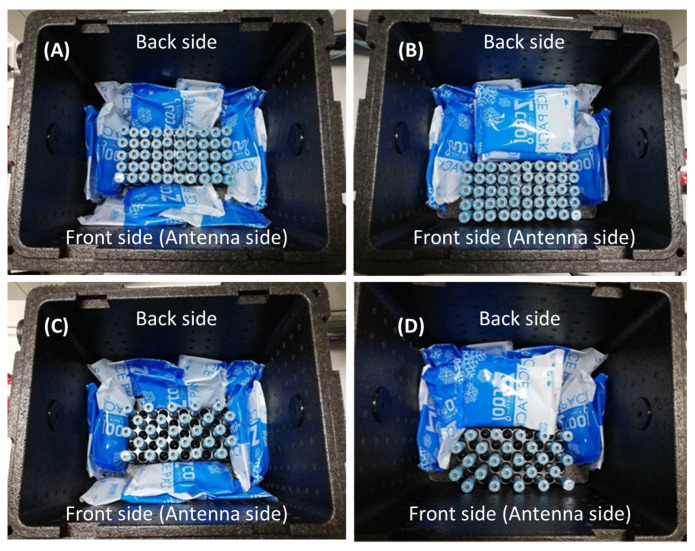
Experiment determining the number of specimen tubes. The tubes are empty. 50 tubes with ice bags surround (**A**), 50 tubes with no ice bags at antenna side (**B**), 25 tubes with with ice bags surround (**C**), and 25 tubes with no ice bags at antenna side (**D**).

**Figure 7 sensors-23-00546-f007:**
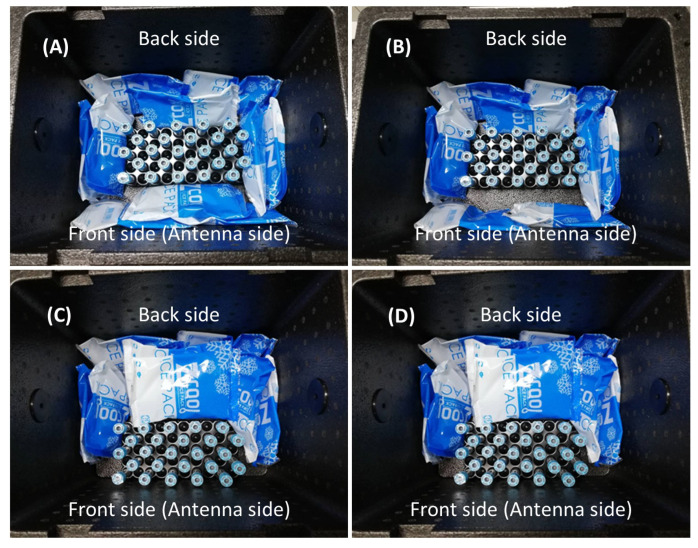
Experiment determining the ice bag’s position inside the transport box. 25 non-liquid tubes and 10 ice bags at 4 sides (**A**), 25 non-liquid tubes and 9 ice bags at 4 sides (**B**), 25 non-liquid tubes and 9 ice bags at 3 sides (**C**), and 25 liquid (water) tubes and 9 ice bags at 3 sides (**D**).

**Figure 8 sensors-23-00546-f008:**
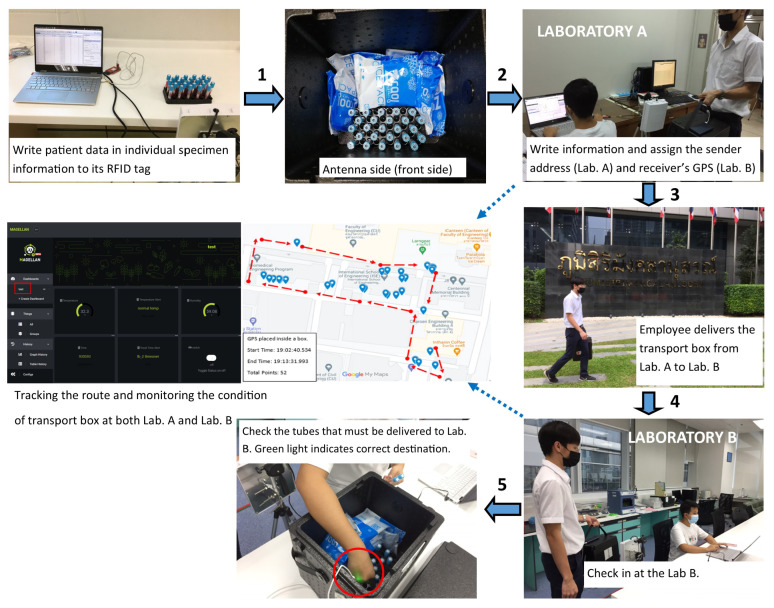
The pilot test of our smart specimen transport system.

**Figure 9 sensors-23-00546-f009:**
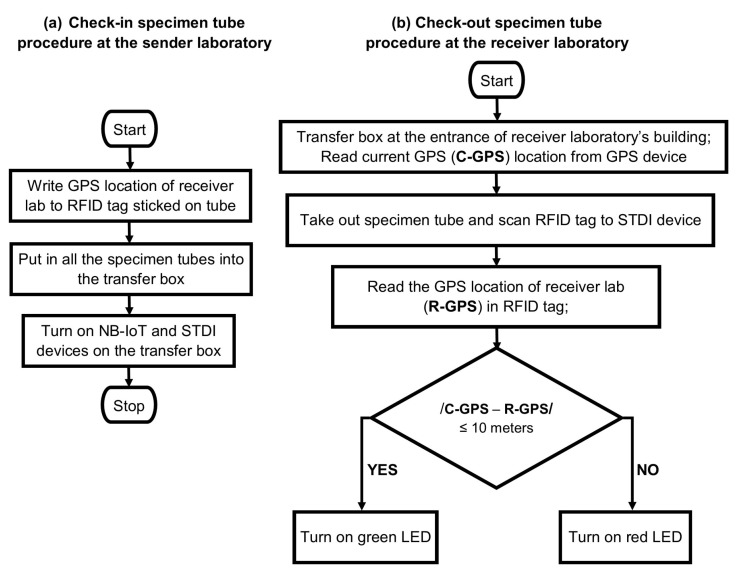
The check-in and check-out specimen tube procedures.

**Figure 10 sensors-23-00546-f010:**
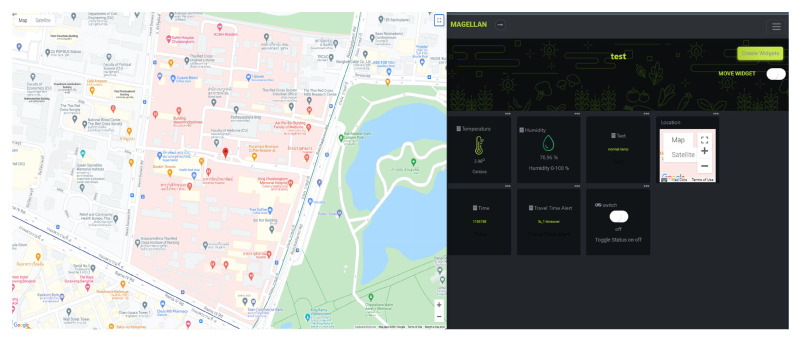
Showing the information of the specimen box during transportation from laboratory A to laboratory B.

**Table 2 sensors-23-00546-t002:** The results of the experiments determining the number of tubes inside the transport box.

Figure	Distance between Box and Antenna (cm)	Number of Empty Tubes	Read Power (dBm)	Number of Detected Tubes (Average)	Percentage
Figure 6A	20	50	27	34	68%
Figure 6B	20	50	25	47	93%
Figure 6C	20	25	27	21	84%
Figure 6D	20	25	25	25	100%

**Table 3 sensors-23-00546-t003:** Module description. The cooler box is made of styrofoam. The RFID read power is 25 dBm.

Figure	Distance between Box and Antenna (cm)	Description	Number of Detected Tubes (Average)
Figure 7A	20	25 non-liquid tubes; 10 ice bags at 4 sides	21
Figure 7B	20	25 non-liquid tubes; 9 ice bags at 4 sides	24
Figure 7C	20	25 non-liquid tubes; 9 ice bags at 3 sides	25
Figure 7D	5	25 liquid (water) tubes; 9 ice bags at 3 sides	25

**Table 4 sensors-23-00546-t004:** Comparison between our study and related studies in the literature.

	Studies	Hun Shim et al. [19]	Kurnianingsih et al. [20]	Kumiko Ohashi et al. [24]	Oscar Urbano et al. [25]	Haider Ali Khan et al. [26]	Sergio Monteleone et al. [27]	Commercial Devices [29,30]	Our System
Features									
Simultaneous RFID read/write		✓	–	–	–	–	–	–	✓
Tracking position		–	✓	✓	✓	✓	✓	–	✓
Remote environment monitoring via IoT		–	–	–	✓	✓	✓	–	✓
Indoor working		✓	✓	✓	–	✓	–	✓	✓
Outdoor working		–	–	–	✓	–	✓	✓	✓

## Data Availability

Not applicable.

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
