# Peer review of "Deployment of Smart Specimen Transport System Using RFID and NB-IoT Technologies for Hospital Laboratory"

_sensors, 2023, doi:10.3390/s23010546_

Round 1

Reviewer 1 Report

The authors of the paper "Deployment of Smart Specimen Transport System using RFID and NB-IoT Technologies for Hospital Laboratory" proposed a system for smart medical items tracking by combination of RFID and an Arduino device for IoT connectivity. The proposed system should ensure the correct delivery of the shipment to the recipient's address and ensure compliance with the storage conditions of the medical items during transport. The value of the article should not be sought in the application of some original idea, but rather in the steps taken in the use of known technologies for the practical solution of a specific problem in medical facilities and detailed elaboration of the implementation. It is also important to note that the proposed technology is part of the transition to digital medicine. The paper is well-written and clearly presented. I have not substantional remarks to the authors.

Author Response

We respect for the reviewer’s time and efforts in reviewing our manuscript.

Reviewer 2 Report

The manuscript (sensors-2077627) is well organized and written as itself though some technical topics havent introduced in detail. For example, how to solve the collision between the RFID tags or the tubes those attached RFID tags? This is very interest to me, but maybe it is beyond the topic of this manuscript. Some other technical details are also interested by readers such as the designs of the transport box and the protocols used in the monitoring system, I also hesitated to give the suggestion about them considering that the manuscript is mainly introduce an application system used in practice based on RFID and NB-IOT.  

Author Response

Author response:  We are thankful for the reviewer’s comment. We would like to response to the reviewer comments as below:

- We addressed the collision between the RFID tags by using the EPC (Electronic Product Code) Gen 2 860-960MHz UHF RFID tag, which supports mutli-read technology. EPC takes full advantage of the higher UHF bandwidth by supporting the reading of multiple tags at once, something not possible in most other RFID technologies. The RFID readers accomplish this by asking tags to randomly select 2 numbers, the reader then decreases the first number incrementally until all tags are read. The second number creates the priority sequence for tags which selected the same first number.

- We used the same design of the transport box which is currently used at the Department of Laboratory Medicine, King Chulalongkorn Memorial Hospital.

- The IoT system is based on AIS NB-IoT board and run the MQTT (Message Queuing Telemetry Transport) protocol for the monitoring.

Author action: In the tracking changes, we revised the manuscript based on the suggestions of reviewer as below:

- We revised the content of the subsection 3.1.1 (page 4) to add more detail about the EPC Gen 2 860-960MHz UHF RFID tag, which supports mutli-read technology.

- We added the detail of the box on the Fig. 4 caption (page 5) in the revised manuscript.

- We added the name of protocol in the revised manuscript (page 6, line 186).

Reviewer 3 Report

1.The paper addresses the following:  (i) an important problem for deployment of Smart Specimen Transport System using RFID and NB-IoT Technologies and (ii) the significant real-world application for Hospital Laboratory (in the Department of Laboratory Medicine, King Chulalongkorn Memorial Hospital).

2.The authors propose a specimen tube prototype and smart specimen transport box using Radio frequency identification (RFID) and Narrow Band – Internet of Things (NB-IoT) technology.

3.The proposed method replaces the existing system, based on barcode technology, with shortage usage and low reliability.

4.In the suggested solution, the passive RFID tag is attached to the surface of the specimen tube and stored information such as patient records, required tests, and receiver laboratory location.

5.The cloud computer system is used as a basis: the environmental condition inside the specimen transport box is sent to the cloud and can be monitored by doctors.

6.The experimental results show the innovation level of the authors method and possibility for integrating RFID and IoT technologies into the specimen logistic system in the hospital.

7.The paper materials are prepared at a very good level (all parts including introduction material, method description, illustrations, experiments description, conclusion and brief description of future research directions, bibliography). The paper will be of interest to readers in various domains (not only in healthcare applications).

8.The paper can be accepted (as is).

Author Response

(The authors gave the same response as above.)
